# Protocol for the Swiss COhort of Healthcare Professionals and Informal CAregivers (SCOHPICA): Professional trajectories, intention to stay in or leave the job and well-being of healthcare professionals

Isabelle Peytremann-Bridevaux [1]*, Vladimir Jolidon [1], Jonathan Jubin [2], Emilie Zuercher [1], Leonard Roth [1], Lucie Escasain[1], Tania Carron[1], Nelly Courvoisier[1], Annie Oulevey Bachmann[2], Ingrid Gilles[3]

1 Center for Primary Care and Public Health (Unisanté), Department of Epidemiology and Health Systems, University of Lausanne, Lausanne, Switzerland, 2 La Source School of Nursing, HES-SO University of Applied Sciences and Arts of Western Switzerland, Lausanne, Switzerland, 3 Lausanne University Hospital, Human Resources Direction, Lausanne, Switzerland

* isabelle.peytremann-bridevaux@unisante.ch

## Abstract

### Introduction

Healthcare professionals' shortage, low job satisfaction, high levels of burnout, and excessive staff turnover are some of the challenges health systems face worldwide. In Switzerland, healthcare stakeholders have called to address the health workforce crisis and have pointed out the scarcity of data on the conditions of healthcare professionals (HCPs). Hence, the Swiss Cohort of Healthcare Professionals and Informal Caregivers (SCOHPICA) was developed to study the career trajectories, well-being, intention to stay in or leave the position/profession/health sector, and their determinants, of HCPs and informal caregivers, respectively. This paper describes the protocol for the HCPs cohort of SCOHPICA and discusses its implications.

### Methods

SCOHPICA is a prospective open cohort using an explanatory sequential mixed methods design. All types of HCPs working directly with patients and practicing in Switzerland are eligible, irrespective of their healthcare setting and employment status. Baseline and annual follow-up electronic surveys will take place once a year, featuring both core questions and modules developed according to information needs. While outcome variables are HCPs' trajectories, well-being, intention to stay in or leave the position/profession/health sector, independent variables include organizational, psychosocial, and psychological determinants, as well as occupational (professional) and sociodemographic factors. The qualitative phase will be organized every two years, inviting participants who agreed to take part in this phase.

**Funding:** The project has been funded by different sources. It has received starting grants from the Swiss Academies of Arts and Sciences, the Swiss Federal Office of Public Health, the Swiss Health Observatory and the Fondation pour l'Université de Lausanne. There is no involvement nor influence of these funders on any stage of the present study, including its protocol, design, data collection and analyses, and results publications and dissemination.

**Competing interests:** The authors declare that they have no competing interests.

The findings from quantitative analyses, along with the issues raised by healthcare stakeholders in the field, will guide the topics investigated in the qualitative phase.

## Discussion

Using innovative methodologies, SCOHPICA will gather nationwide and longitudinal data on HCPs practicing in Switzerland. These data could have numerous implications: promoting the development of research related to HCPs' well-being and retention intentions; supporting the development of policies to improve working conditions and career prospects; contributing to the evolution of training curricula for future or current healthcare professionals; aiding in the development of health systems capable of delivering quality care; and finally, providing the general public and stakeholders with free and open access to the study results through an online dashboard.

## 1. Introduction

Healthcare professionals (HCPs) are the cornerstone of health systems. Achieving high performing health systems and the highest standards of healthcare requires a sufficient supply of HCPs who are well-trained, adequately distributed throughout the system, and who have appropriate working conditions to be able to provide care that is accessible, equitable and of good quality [1, 2]. In this sense, the World Health Organization (WHO) sets the following health workforce objectives for 2030: (1) to increase the performance of the health workforce; (2) to match investments in the health workforce to population needs; (3) to build health institution capacity at all levels; (4) and to reinforce data collection on the health workforce [2].

European countries are facing considerable challenges with their health and care workforce. A recent WHO report highlighted personnel shortages, insufficient recruitment and retention, migration of qualified workers, unattractive working conditions, and poor access to continuing professional development opportunities, as current issues which are threatening health systems [1]. The report also stressed the poor mental health of the workforce across Europe, related to long working hours, inadequate professional support and staff shortages. Several of these problems were exacerbated during the COVID-19 pandemic, as healthcare systems suffered high pressure and the workforce had to cope with increased workloads, job-related stress, and physical and mental health risks [3]. The issue of HCPs' retention has been a concern for over a decade [4]. Since 2012, the European Union has launched several programs, such as the Join Action Plan [5] and the Support for the Health Workforce Planning and Forecasting Expert Network, [6] aimed at improving the retention of HCPs and addressing their shortage. In Switzerland, efforts have been made by health departments of the universities of applied sciences, which collaborated to create the Competence Network Health Workforce [7]. This network aims to define a national strategy to tackle shortages of HCPs. At the political level, the Swiss population approved a constitutional law in 2021, compelling Swiss cantons and the federal government to ensure the sufficient availability of qualified nurses, and therefore to collect data to monitor the implementation of this new law [8]. Despite these actions, data and analytical capacity are lacking, which jeopardizes the strategic planning of the health workforce [1]. In sum, data is currently not sufficient to address existing challenges, and to effectively plan, manage, coordinate and inform decisions on the health workforce.

Health workforce research has covered various areas, mainly relating to HCP's physical and mental health [9–13], and aspects related to the functioning of the healthcare system, such as absenteeism, career changes, figures of HCPs who are employed or undertaking training (and their projections), turnover and intention to stay in/leave their position/profession [14–18]. Four areas require further investigation, nevertheless. Firstly, more in-depth analyses are needed on the interconnections, causal relations and mediating pathways between the determinants of HCPs' well-being and intention to stay in/leave the position/profession/health sector. That is, to date, research has mainly focused on the individual role of organizational (e.g., recognition, leadership, work environment, workload), psychosocial (e.g., cohesion and social support at the workplace), psychological (e.g., stress, resilience, engagement), and sociodemographic (e.g., age, gender, seniority) determinants [19–22]. Secondly, the professional trajectories of HCPs, from initial training to retirement, remain understudied. Currently, most studies have relied on cross-sectional study designs which do not fully capture the longitudinal experience of HCPs. Also, to our knowledge, life history calendar (LHC), a tool designed to collect retrospective data from participants by maximizing their possibilities of recalling past events completely and accurately [23, 24], has not been used in health workforce research yet. LHCs have proven successful across various contexts, including studies on the trajectories of unemployed and vulnerable individuals, the sexual life of young people [25, 26], and in general population surveys [27]. These studies have shown that 1) LHCs are more efficient than traditional sociodemographic questions for collecting retrospective data; 2) the data collected is reliable [28]; and 3) online versions of LHCs can be used to reach large samples of participants [29, 30]. Using LHC could help understand HCPs' career trajectories thoroughly, from their training to their current situation, and provide a typology of professional trajectories. Concerning past cohort studies on HCPs, only a limited number of these studies have explored multiple healthcare professions, and these did not delve into professional trajectories of HCPs or their relation to HCPs' well-being and intention to stay in/leave their job [31–36]. Given their suitability to study the dynamics of ever-changing health workforce markets, it is particularly appropriate to consider cohort studies following participants over time and allowing for the monitoring of their trajectories [37]. Thirdly, despite a wealth of literature on HCPs, it has mainly focused on physicians or nurses and other healthcare professions have been understudied [1, 38], and few studies covered a variety of healthcare professions [37, 39, 40]. Importantly, as a recent review pointed out [41], issues of well-being and intentions to leave the profession have affected HCPs other than physicians and nurses, yet these professions have received far less attention. Finally, it is key to investigate settings beyond the two most frequently studied, namely hospitals and general practices.

In Switzerland, the deteriorating working conditions of HCPs and staff shortages have been stressed by scientific studies and reports for several years, and this situation has worsened since the COVID-19 pandemic. Reports from the Swiss Health Observatory have predicted that a large number of HCPs would need to be hired to meet population needs, and that physicians' supply in the ambulatory sector would not be sufficient by 2030 [42–45]. Additionally, a recent report has highlighted that 70,000 nursing staff will be needed by 2029, which encompasses both workforce replacements needs and the increased demand for additional staff stemming from population healthcare needs [46]. In fact, the coverage rate is predicted to be lower than 80% with a clear deficit between workforce supply and projected needs [46, 47]. This situation mirrors an international trend. Indeed, the WHO has projected a shortage of 15 million HCPs by 2030. In Germany, for example, estimates for the required number of HCPs in 2030 ranged from approximately 263,000 to nearly 500,000 full-time equivalents [48]. Similarly, as of September 2023, the UK's National Health Service (NHS) reported 121,000 full-time equivalent vacant positions [49]. Finally, several reports have indicated that the United States will

face a shortage of up to 124,000 physicians by 2033 and will require 200,000 nurses annually to meet the increasing care demand [50]. As in other countries, Swiss healthcare stakeholders have stressed the paucity of data, hindering effective monitoring, planning, and managing of the health workforce. Research projects aimed at both collecting data and leveraging HCPs retention have also been conducted in the Swiss context. These have investigated job stress, job satisfaction, burnout, and intention to leave the job/profession [51–65]. However, like studies conducted in other countries, these publications mostly concentrated on nurses and physicians (mainly in hospital setting), and both nationwide and longitudinal data across multiple healthcare sectors are lacking to understand HCPs' professional trajectories, well-being, and intention to stay in/leave their position/profession/health sector.

In this international and Swiss context, we developed the Swiss COhort of Healthcare Professionals and Informal CAregivers (SCOHPICA) to collect nationwide and longitudinal data to better understand the trajectories and work experience of HCPs, and help to tackle the Swiss health workforce crisis. Considering mixed methods and using both quantitative and qualitative data, the HCPs cohort of SCOHPICA aims at 1) investigating the professional trajectories of HCPs from the completion of their training onwards; 2) examining the intention to stay in or leave the position/profession/health sector, well-being, and their determinants; 3) providing an in-depth understanding of the mechanisms leading HCPs to stay in/leave the position/profession/health sector; 4) making the data and results available to all healthcare stakeholders, researchers and the public in general, through a secured data repository and an online interactive platform. The SCOHPICA project is conducted by an interdisciplinary team based at Unisanté, La Source School of Nursing–University of Applied Sciences and Arts (HES-SO), and Lausanne University Hospital, all located in Lausanne, Switzerland.

## 2. Methods

### 2.1. Study design

SCOHPICA is a national prospective open cohort that uses an explanatory sequential mixed methods design, initially collecting quantitative data and subsequently explaining the quantitative results with in-depth qualitative data, to foster a more comprehensive and complete understanding of specific research questions [66]. While the longitudinal design will collect essential quantitative data on HCPs, the qualitative phase will provide in-depth analyses of issues identified in the quantitative phase.

Although SCOHPICA aims to study both HCPs and informal caregivers (ICs), who are key but often neglected actors of the health system, the present protocol focuses on HCPs since the implementation of SCOHPICA's informal caregivers' cohort will start in the Spring of 2024 (the protocol for the ICs part of SCOHPICA will be published separately).

### 2.2. Population and setting

All HCPs (e.g., general practitioners, specialist physicians, nurses, nurse aides, paramedics, medical assistants, pharmacists, physiotherapists, psychologists, dieticians, etc.) working directly with patients and currently practicing in Switzerland, irrespective of the setting (e.g., hospitals, clinics, nursing homes, private practices, community services) and their employment status (e.g., self-employed, salaried), are eligible to participate in SCOHPICA. Students and HCPs who left their profession or are retired, at the time of the baseline survey, are not eligible to participate in the study. However, participants who leave their job or their profession after having joined SCOHPICA will be retained in the cohort. Finally, HCPs who are unable to read any of the Swiss national languages (French, German and Italian) are not eligible.

**2.2.1. Sample size.** In the quantitative phase, SCOHPICA aims to collect data from 5,000 to 10,000 unique baseline HCPs, whom we will follow over the years. The latter sample size was estimated to achieve satisfactory measurement precision around the outcome variables, and sufficient power for global cross-sectional and longitudinal analyses, as well as cluster analysis of professional trajectories. Sample size calculations considered the expected values (and variability) of outcome variables as reported in previous research, a type I error of $\alpha = 0.05$ (two-sided) and 95% confidence intervals around the possible values of outcome variables.

Since SCOHPICA is an open cohort, new participants will be recruited every year between October 1$^{st}$ and January 31 of the following year. This will contribute to increasing the number of participants and improving the statistical power needed to conduct relevant sub-group and stratified analyses.

For the qualitative phase, participants who agreed to be contacted in the baseline survey will be invited for individual or group sessions. Every two years, we aim to conduct about 15 group sessions, each consisting of eight HCPs, totaling 120 participants. This target may be adapted according to the characteristics of participants, the chosen topic, the specific method [67] and data saturation assessment [68]. Participants will be invited based on the topics that need to be deepened, particularly those stemming from the analyses of previously collected quantitative data. We will use purposive sampling and will aim to obtain a heterogeneous sample, covering different sociodemographic profiles, linguistic regions, sectors of activity (e.g., hospital sector, private practices), as well as professions that have been affected by personnel shortage (e.g., nurses, general practitioners, pharmacy assistants). Since a second aim of the qualitative part will be to explore specific professional trajectories, participants will also be selected according to profiles of trajectories emerging from the quantitative analyses.

**2.2.2. Recruitment of healthcare professionals.** Due to the absence of comprehensive records nor registries of all HCPs practicing in Switzerland (i.e., providing access to HCPs' contact details), multiple communication and recruitment strategies are used to reach HCPs through different organizations. Professional, state and umbrella associations of all types of HCPs at national, regional and cantonal levels (i.e., the 26 administrative divisions of Switzerland), as well as HCPs employers (e.g., hospitals, home care), are contacted to request their support in recruiting their members. Communication packages are created for recruitment purpose and provided to these entities so they may share SCOHPICA's information, website and electronic questionnaire link with their members, for example through ad-hoc emails, newsletters and their own websites. Finally, social media are also used (our own institutional platforms and those of organizations promoting the recruitment). Records of the contact details of all the organizations and individuals who are contacted, and those who are willing to support the recruitment process, are kept for future annual recruitments.

Additionally, short articles are published in the journals of professional associations, the project is presented at large conferences in Switzerland, and a kick-off meeting including a press conference was held in September 2022. To promote awareness of SCOHPICA among stakeholders and the general public, a series of conferences and events focused on the health workforce are being organized and conducted.

## 2.3. Data collection and measures

SCOHPICA comprises quantitative and qualitative data collection phases, as detailed in the sections below and summarized in Fig 1.

**2.3.1. Quantitative phase.** *Baseline survey*. Baseline data is collected using a self-reported electronic questionnaire accessible to all HCPs working in Switzerland on SCOHPICA website

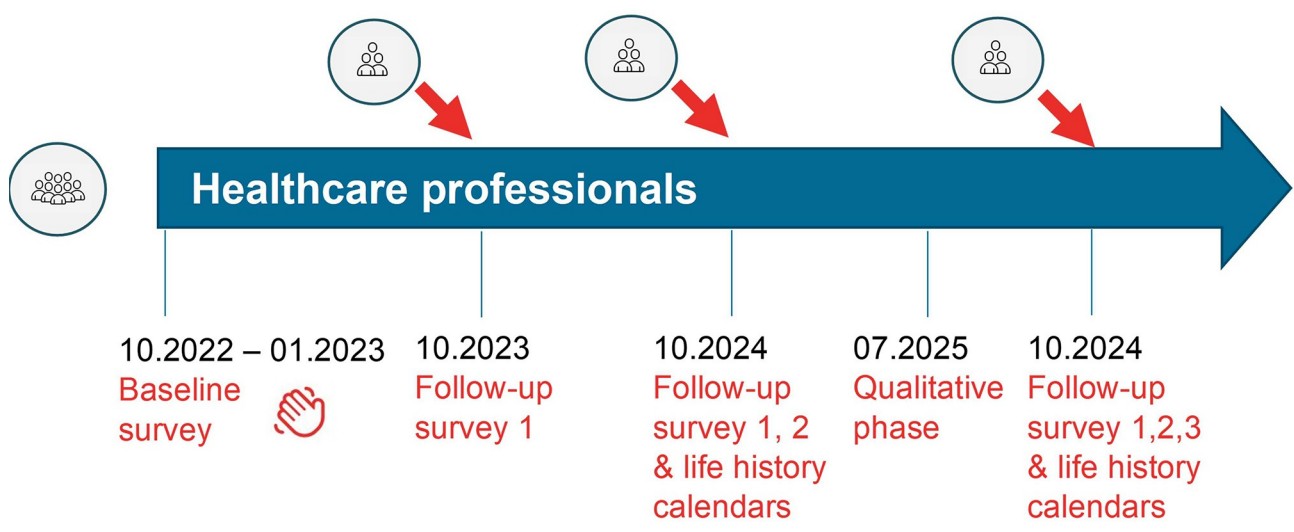

**Fig 1. Data collection flowchart.**

(www.scohpica.ch). The questionnaire contains ~140 questions, including three open-ended questions on professional aspects, and takes approximately 30 minutes to complete. Originally developed for HCPs working in hospitals and health institutions, it is slightly adapted for HCPs working in private practices. At the end of the questionnaire, participants are asked to provide their email address if they wish to be contacted for follow-up surveys; they are also asked whether they agree to be contacted to participate in SCOHPICA's subsequent qualitative phase (i.e., individual and group sessions). SCOHPICA first baseline survey took place between October 1, 2022 and January 31, 2023.

*Outcome variables.* The main outcomes of SCOHPICA are:

- Professional trajectories, created based on socio-professional information (from baseline, LHC and follow-up surveys)

- Intention to stay in the position / profession / health sector (3 items; 5-point scale from *No, not at all* to *Yes, absolutely*)

- Intention to leave the position / profession / health sector (3 items; 5-point scale from *Very unlikely* to *Very likely*)

- Well-being, assessed with the Flourish Index (10 items; 10-point scale from *Extremely unhappy* to *Extremely happy*) [56]

*Independent variables.* Determinants of the intent to stay/leave and well-being. The questionnaire collects data on the determinants of the above-mentioned outcomes, as presented in Table 1. These determinants were selected based on preliminary systematic reviews targeting the nursing professions, physicians and allied HCPs [22, 41], as well as on discussions with experts from the SCOHPICAs' national support panel, including the Swiss Federal Office of Public Health, HCPs Swiss platform, HCPs' associations, representatives of universities and universities of applied sciences. This allowed for identifying the most important determinants affecting HCPs' well-being and intentions to stay in/leave the position/profession/health sector. We chose the following validated scales to measure these determinants (see details in Table 1):

- Quantitative workload inventory [69]: workload.

**Table 1. Main determinants measured in SCOHPICA's questionnaire.**

| Dimension | Instrument and source | Measurement (number of items and scale) |
|---|---|---|
| Workload | Quantitative workload inventory [69] | 5 items; 5-point scale from *Less than once a month/Never* to *Several times a day* |
| Control over working time | COPSOQ [70] | 5 items; 5-point scale from *Never/Hardly ever* to *Very often/Always* |
| Staffing and resource adequacy | PES-NWI [71] | 5 items; 4-point scale from *Strongly disagree* to *Strongly agree* |
| Possibilities for development | COPSOQ [70] | 3 items; 5-point scale from *To a very large extent* to *To a very small extent* |
| Work-life conflict | COPSOQ [70] | 5 items; 4-point scale from *Yes, absolutely* to *No, not at all* |
| Leadership | Global transformational leadership scale [72] | 7 items; 5-point scale from *Never/Hardly ever* to *Very often/Always* |
| Influence at work | COPSOQ [70] | 6 items; 5-point scale from *Never/Hardly ever* to *Very often/Always* |
| Sense of community at work | COPSOQ [70] | 3 items; 5-point scale from *Never/Hardly ever* to *Very often/Always* |
| Interprofessional collaboration | Intensity of interprofessional collaboration [73] | 6 items; 5-point scale from *Strongly disagree* to *Strongly agree* |
| Recognition at work | Recognition at work scale [74] | 12 items; 5-point scale from *Strongly disagree* to *Strongly agree* |
| Preparedness to work reality | No instrument available—created by the research team and expert advisory group | 2 items; 5-point scale from *Strongly disagree* to *Strongly agree* |
| Meaning of work | COPSOQ [70] | 2 items; 5-point scale from *To a very large extent* to *To a very small extent* |
| Moral resilience | Rushton moral resilience scale [75] | 9 items; 4-point scale from *Disagree* to *Agree* |
| Intolerance to uncertainty | Intolerance of uncertainty scale [76] | 6 items; 5-point scale from *Not at all my characteristic* to *Entirely my characteristic* |
| Burnout | The one-item MBI:EE [77] | 1 item; 5-point scale from *I do not have burnout symptoms* to *I feel completely burned out* |
| Self-rated health | SF36 [78] | 1 item; 5-point scale from *Poor* to *Excellent* |
| Job satisfaction | COPSOQ [70] | 1 item; 4-point scale from *Very unsatisfied* to *Very unsatisfied* |
| Quality of care | Adapted from Aiken et al. [79] and Shanafelt et al. [80] | 9 items; 5-point scale from *Strongly disagree* to *Strongly agree*<br>1 item; 5-point scale from *Poor* to *Excellent*<br>4 item; 5-point scale from *Never* to *Every week* |

Note: Answer format for item measurements were Likert scales.

Acronyms: COPSOQ: Copenhagen Psychosocial Questionnaire; PES-NWI: Practice Environment Scale of the Nursing Work Index; SF36: 36-Item Short Form Survey

- Copenhagen Psychosocial Questionnaire (COPSOQ; [70]): control over working time; possibilities for development; work-life conflict; influence at work; sense of community at work; meaning of work; job satisfaction.

- Practice Environment Scale of the Nursing Work Index (PES-NWI; [71]): staffing and resources.

- Global transformational leadership scale [72]: transformational leadership.

- Intensity of interprofessional collaboration [73]: interprofessional practice.

- Recognition at work scale [74]: recognition from managers, colleagues and patients.

- Rushton moral resilience scale [75].

- Intolerance of uncertainty scale [76].

- The one-item MBI:EE [77]: non-proprietary single-item of burnout.

- Self-rated health single item [78].

- Organizational Support Subscale of the Nursing Work Index [79]: perceived quality of care.

Four none-validated items were drawn from a study by Shanafelt and colleagues [80] to complement the assessment of perceived quality of care. Moreover, two questions were developed by the research team to assess "Preparedness to work reality", as no validated scale is currently available for this construct. We conducted a qualitative pre-test of the questionnaire with 20 participants from diverse backgrounds.

Additionally, the baseline questionnaire collects information on respondents' work and occupation-related aspects (Table 2). Socio-professional information included: employment status, professional status, employment rate, occupational context, current profession, training and specialization, additional training, years of professional experience, main occupational domain, hours worked per day, days worked per week, type of work shift, type of work schedule, managerial responsibilities, country of education, other past profession(s), modification of employment rate in past 12 months, number of employers/occupational context/job location changes since starting to work in healthcare, career interruption(s) of one year or longer, unemployment period(s), work-related accident(s)/sick leave(s), healthcare sector throughout career, healthcare sector of first employment, work location, and commuting time to work. Sociodemographic information included: gender, year of birth, nationality, marital/partnership status, children, informal caregiving status, monthly individual incomes, monthly household incomes, and residency location.

*Follow-up surveys*. Follow-up surveys will be implemented annually using a questionnaire that includes an unchanged core set of questions across all survey waves. Additionally, modules of questions may be added or removed based on the HCPs' current situation, emerging priority topics and results of previous surveys.

In fall 2024, the follow-up survey will include an online life history calendar (LHC), a tool allowing the retrospective and accurate collection of data on individuals' past professional and non-professional events [23, 24]. Respondents will be invited to complete the LHC in the first or second year after completing the baseline questionnaire.

**2.3.2. Qualitative phase.**   Respondents who accepted to be contacted in the baseline survey will be invited for individual or group sessions which will take place every two years. These will last about 60 to 90 minutes.

Topics to be investigated in individual and group sessions will emerge from the annual quantitative analyses, as well as from the needs and issues reported by healthcare stakeholders and experts in the field over time. These will be selected to deepen the understanding of (among other topics): 1) the interrelations between the determinants of the intention to stay in/leave the position/profession/health sector, and well-being; 2) the specific profiles of professional trajectories emerging from quantitative analyses (and their relation to the intention to stay in/leave the position/profession/health sector); and 3) the profiles of those HCPs who actually left their position/profession/health sector.

The qualitative method approaches used in the project will vary according to the specific topics under investigation. However, the project will mainly use the phenomenological [81, 82] and concept mapping [83] approaches. These two methods are particularly appropriate: 1) to understand how individuals experience and construct meaning on phenomenon they experience using both their inner perceptions and environment appraisal; 2) to construct a collective and structured representation of a topic that produces "an interpretable pictorial view of ideas and concepts and how these are interrelated" [84]. Concept mapping provides a visual representation of complex data. It allows for synthesizing complex qualitative information in a relatively short timeframe, making it easier to identify relationships and emergent themes.

**Table 2. Socio-professional and sociodemographic questions included in SCOHPICA's questionnaire.**

| Questions | Measurement (number of items and scale) |
|---|---|
| Socio-professional questions: | |
| Employment status | Employed, seeking employment, non-employed due to disability, homemaker, retired, undertaking training/student/apprentice, undertaking retraining for another profession, unemployed and not seeking employment, other |
| Professional status | Employee, self-employed, manager/administrator. |
| Employment rate | From 0% to 100% |
| Occupational context | Public hospital, private hospital, solo/two-physician practice, group practice, home care, nursing home, pharmacy, etc. |
| Current profession | Paramedic, physician, medical assistant, pharmacist, midwife, registered nurse, physiotherapist, etc. |
| Training and specialization | For nurses and physicians |
| Additional training | Yes, no, if yes then open-ended answer |
| Years of professional experience | Number of years |
| Main occupation domain | Somatic care, home care, mental health, rehabilitation, long-term care, other |
| Hours worked per day | Number of hours |
| Days worked per week | Number of days |
| Type of work shift | Day only, night only, both day and night |
| Type of work schedule | Continuous, interrupted |
| Managerial responsibilities | Yes, no, if yes then number of employees managed |
| Country of education | Open-ended answer |
| Other past profession(s) | Yes, no, if yes then open-ended answer |
| Modification of employment rate in past 12 months | Yes, no, if yes then increased or decreased, and reason for |
| Number of employers/occupational context/job location changes | Number |
| Since starting to work in healthcare, career interruption of one year or longer | Yes once, yes several times, never, if yes then number of interruptions and reason for interruption and return (open-ended answer) |
| Unemployment period | Yes once, yes several times, never |
| Work-related accident(s)/sick leave(s) | Yes, no |
| Healthcare sector throughout career | Public sector only, mainly public sector, private sector only, etc. |
| Healthcare sector of first employment | Public sector, private sector, broader public sector |
| Work location | Canton |
| Commuting time to work | Hours and minutes |
| Sociodemographic questions: | |
| Gender | Man, woman, other, do not wish to answer |
| Year of birth | Year |
| Nationality | Swiss, Swiss and other nationality, foreign national |
| Marital/partnership status | Cohabiting partner/registered partnership/married, separated/dissolved partnership/divorced, single, widowed |
| Children | Number of children and age range |
| Informal caregiving status | Yes currently, yes in the past, no |
| Monthly individual incomes | Below 2000, 2001–4000, 4001–6000, 6001–8000, 8001–10000, more than 10000 CHF |
| Monthly household incomes | Below 3000, 3001–6000, 6001–9000, 9001–12000, 12001–15000, more than 15000 CHF |
| Residency | Name of the canton |

In line with phenomenological and concept mapping approaches, we will conduct individual or group sessions, including one-to-one interviews. The use of individual or group sessions will be guided by the specific methodology underlying each approach (i.e. the 6 steps of concept mapping, alternating brainstorming and individual sessions) or by the nature of the topics under investigation. For example, topics such as interprofessional collaboration are well-suited to group sessions which favor the sharing of experiences and facilitate the examination of contexts that promote well-being and retention. Other topics which are more sensitive and personal, such as burnout, may be better explored through one-to-one interviews.

## 2.4. Data management

The software Le Sphinx iQ2 and Le Sphinx iQ3 are used for the survey design, data collection and generation of databases. Databases will be converted to R/SPSS/Stata formats for analyses. Individual and group sessions will be recorded and integrally transcribed, and audio recordings will be deleted. To preserve respondents' confidentiality, their names will not appear in databases (nor in interview transcripts) used for analyses. All data will be stored in a secured institutional server. The project complies with the General Data Protection Regulation (GPRD) as well as the requirements from the Cantonal Research Ethics Committee, Vaud (CER-VD).

## 2.5. Data analysis

**2.5.1. Quantitative phase.** First, descriptive analyses will summarize the distributions of each variable, including patterns of missing data. The internal consistency of score variables will be evaluated with Cronbach's alpha. Then, bivariate analyses will be conducted, which will inform the development of multivariate regression models to adjust for potential confounding factors. Mixed-effects models will be considered to account for data dependency structures. The effects of organizational, psychosocial and sociodemographic factors on well-being and intention to stay in/leave the position/profession/health sector will be examined with regression analyses, and structural equation modelling (SEM) will be used to assess mediating pathways and causal relations.

Additionally, we will perform sequence analysis with LHC data [85]. Specifically, we will apply optimal matching and clustering techniques to construct a typology of HCP's professional trajectories. Then, the longitudinal profiles thus identified will be related to the sociodemographic information and determinants from the baseline survey.

Sample characteristics such as age, gender and professional group will be systematically compared with available national statistics, and weighted analyses will be considered in case of structural discrepancies.

The default approach to address missing observations will be to either replace nonresponses with informed choices of values wherever possible or proceed with multiple imputation. Listwise deletion will only be considered in specific cases, such as univariate analyses. For participants with missing answers in items of dimensions (instruments), their scores will be calculated based on the mean of the items to which they answered, provided they answered more than 50% of the items and at least two items within the dimension, and if the dimension's internal consistency proved to be acceptable.

All quantitative analyses will be performed using R (Stata, SPSS or another specific statistical software may also be used).

**2.5.2. Qualitative phase.** For the qualitative data analysis, the analytical strategy will depend on the specific qualitative approach. In the case of "concept mapping", a participatory research method in public health described by Burke et al. [83] and based on the work of

Trochim [84], researchers engage participants in the process of creating concept maps, allowing them to visually represent their thoughts. In this framework, participants are involved in collaborative mapping sessions, which not only enable a deeper exploration of their perspectives but also actively involve them in the data collection and analysis process [86]. Concept mapping is divided into six steps: 1) Preparation; 2) Generation, 3) Structuring, 4) Representation, 5) Interpretation, and 6) Utilization, which help to ensure a rigorous and systematic approach to data analysis and interpretation.

Concerning the phenomenological approach, data from individual and group sessions will be fully transcribed and analyzed following the 6 steps process recommended by Smith and colleagues [87]: 1) full reading of transcripts, 2) first annotations, 3) emergent themes identification, 4) construction of links between themes, 5) iterating process among all the cases (i.e. individual or group sessions), 6) identification of links between cases.

To integrate qualitative and quantitative data into our analysis, we will follow Creswell and Plano's Joint Display technique [66]. To implement this technique, we will use the Pillar Integration Process (PIP), a systematic process of mixed-methods data integration consisting of four steps: Listing, Matching, Checking and Pillar Building [88].

MAXQDA or ATLAS.ti softwares will be used for qualitative analyses, as well as IRaMuTeQ for textual data analysis (i.e., to analyze open-ended questionnaire responses).

## 2.6. Ethical considerations

SCOHPICA is an observational cohort study which does not expose participants (i.e., HCPs) to health risks. In the surveys, and individual and groups sessions, respondents participate upon their free will and without compensation, and they can withdraw from participation at any time. At baseline, before accessing the electronic questionnaire, participants are directed to a consent form where they indicate, first whether they agree to take part in the study under the conditions outlined in an information sheet, and then, whether they consent to the use of their data in future studies. They are then directed to the questionnaire by clicking "yes", while clicking "no" to the first consent closes the questionnaire. Individuals' data is coded so respondents cannot be identified without the corresponding key, and data is handled with strict adherence to confidentiality standards. This double consent is valid for the quantitative part of SCOHPICA; for the qualitative part, a specific informed consent will be required.

Although this research project may not directly benefit research participants, it has the potential to indirectly benefit them and other HCPs in the future. Namely, this project primarily holds social value because by offering insights into the determinants of HCPs' well-being and trajectories, it will contribute to informing strategic political and institutional decisions concerning the health workforce, such as interventions aimed at improving HCPs retention and working conditions.

Ethical approval was obtained from the Cantonal Research Ethics Committee, Vaud (CER-VD), Switzerland (project ID: 2022–01410) and the project was registered on ClinicalTrials.gov, identifier: NCT05571488. A separate ethics committee approval will be necessary for the qualitative phases of SCOHPICA.

## 3. Results and data dissemination

Since SCOHPICA data and results are essential for decision-making and research in healthcare, they will be widely and publicly disseminated. Firstly, results will be reported and accessible on an online interactive platform (i.e. dashboard) with data visualization tools such as reports, indicators, tables, and charts. Secondly, SCOHPICA de-identified datasets and metadata will be made available upon request through a secured data repository in accordance with

the FAIR principles. Finally, dissemination of study results will also be done through scientific peer-reviewed and lay publications, as well as presentations at international, national and regional conferences, reaching both scientific and non-scientific audiences.

## 4. Discussion

The HCPs part of SCOHPICA described in this protocol is an ambitious and innovative project which collects nationwide and longitudinal data among healthcare professionals (HCPs) to better understand their professional trajectories, work conditions and experiences, and well-being. This project has unique characteristics which will contribute to both academic research and policy-making in the field of health workforce. It is highly relevant for the international research community as it contributes to understudied health services research areas, using both quantitative and qualitative data. Firstly, this project covers all types of practicing HCPs, whereas previous research did not have such a comprehensive coverage of health professions, allowing to compare their diverse experiences and conditions. Secondly, it applies a longitudinal approach, incorporating both a cohort design and the use of LHC, an original trajectory data collection method not previously used among HCPs. These unique and innovative approaches will provide new evidence on the multiple determinants affecting the health workforce. Combined with advanced statistical techniques, such as structural equation modelling, this will help to uncover how the interconnections and pathways between these determinants shape professional trajectories, intentions to stay in/leave the position/profession/health sector, and well-being of HCPs. Thirdly, this project is important for Swiss healthcare stakeholders who have stressed the lack of data on all types of HCPs in Switzerland, which is crucial for addressing healthcare system issues. If actions are not taken, inadequate workforce planning may exacerbate attrition and burnout, resulting in staff shortages, increased workloads and greater difficulties in organizing healthcare [89, 90]. Thus, SCOHPICA's results and data are key to design management and policy interventions aimed at improving the health workforce conditions and retaining HCPs. In this regard, the Swiss Federal Office of Public Health has commissioned SCOHPICA with providing indicators for monitoring the conditions of nursing and care staff, beginning in 2024. Moreover, several Departments of Health of Swiss cantons have requested specific reporting concerning their local health workforce. Fourthly, SCOHPICA's results dissemination strategy goes beyond traditional methods since the results will be published on an online interactive platform with data visualization options allowing users to customize the reporting. Additionally, the data will be available on a data repository, which will facilitate research collaborations and support policy-making addressing key issues related to the health workforce. Finally, thanks to a grant from the Fondation pour l'Université de Lausanne, the dissemination of results to all stakeholders and civil society, through public forums and consensus forums, will be made possible.

Apart from the above-mentioned strengths of SCOHPICA, its limitations should be considered. First, non-probability sampling is used as it is not feasible to draw representative samples of all HCPs and obtain their contact emails in the current Swiss context. To assess the representativeness of our sample of participants, we will compare their socio-demographic characteristics (i.e., gender, age and professional groups) with relevant national statistics, data from professional associations or Swiss published studies, where available. Second, some HCPs subgroups may be under-represented during the initial years of the project, such as specific professional categories that are more challenging to recruit. With the successful demonstration of SCOHPICA's feasibility through the first recruitment, we are confident that participation from the various professions will rapidly increase in the future. Third, as in studies using a similar methodology, there is a risk of selection bias if individuals who do not respond to the

survey systematically differ in their individual characteristics from the respondents. Since we will not have information on non-respondents, it will not be possible to compare their characteristics with those of respondents. Yet, we expect that yearly recruitments will help increase SCOHPICA's sample size and better represent the Swiss health workforce. Forth, SCOHPICA relies on self-reported data, which may be susceptible to recall and social desirability biases, potentially introducing measurement bias. To reduce such bias, SCOHPICA questionnaire relied on widely used and validated questions, and the questionnaire was pre-tested in the three national languages. Finally, participants may drop out or become lost to follow-up during the course of the study. This attrition phenomenon can introduce bias if the characteristics of those who drop differ from those who remain in the study. We will assess this bias by comparing key characteristics of respondents who quit and those who remain in SCOHPICA. Additionally, we will mitigate participant dropout by making it easy and motivating for them to participate and stay in the study. For instance, we will present results and show how they are used by stakeholders, make the data and results available to all, send New Year cards to participants, minimize the frequency of contacts and follow-ups, and maintain up-to-date contact information to sustain communication, even if participants relocate.

To conclude, SCOHPICA targets understudied areas in the health workforce domain, filling knowledge gaps and addressing existing limitations. It will provide relevant data and evidence by studying all types of HCPs practicing in Switzerland, considering the various determinants of professional trajectories, intention to stay in or leave the position/profession/ healthcare sector, and well-being. Facilitating access to data and results will be particularly valuable to national and international healthcare stakeholders and researchers. By supporting the monitoring, planning, and management of the Swiss health workforce, SCOHPICA will be key for tackling health system challenges, designing future policies, implementing ad hoc interventions and promoting the delivery of high quality of care.

## Acknowledgments

The conceptualization and launch of this project would not have been possible without the support of partners from different institutions, administrative and communication staff, part-time assistants and interns from Unisanté, the Institut et Haute Ecole de la Santé La Source and the Lausanne University Hospital, scientific experts and SCOHPICA's advisory panel including Aide et soins à domicile Suisse, ARTISET, Association suisse des infirmières et infirmiers, Competence Network Health Workforce, la Conférence des directrices et directeurs cantonaux de la santé, Médecins de famille et de l'enfance Suisse, pharmaSuisse, Swiss Federal Office of Public Health, Swiss Health Observatory, the Swiss Medical Association, Swiss Nurse Leaders, Unisanté and University of Lausanne. We also would like to thank all individuals and the professional, state and umbrella associations of HCPs at national and cantonal level who supported SCOHPICA's first 2022 recruitment phase of HCPs). Last but not least, we would like to thank all the participants for contributing their time and participating in this study.

## Author Contributions

**Conceptualization:** Isabelle Peytremann-Bridevaux, Nelly Courvoisier, Annie Oulevey Bachmann, Ingrid Gilles.

**Funding acquisition:** Isabelle Peytremann-Bridevaux.

**Methodology:** Isabelle Peytremann-Bridevaux, Jonathan Jubin, Leonard Roth, Nelly Courvoisier, Annie Oulevey Bachmann, Ingrid Gilles.

**Project administration:** Isabelle Peytremann-Bridevaux.

**Supervision:** Isabelle Peytremann-Bridevaux, Annie Oulevey Bachmann, Ingrid Gilles.

**Writing – original draft:** Isabelle Peytremann-Bridevaux, Vladimir Jolidon.

**Writing – review & editing:** Isabelle Peytremann-Bridevaux, Vladimir Jolidon, Jonathan Jubin, Emilie Zuercher, Leonard Roth, Lucie Escasain, Tania Carron, Nelly Courvoisier, Annie Oulevey Bachmann, Ingrid Gilles.

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
