## [Decision Letter · Decision Letter 0]

27 Mar 2024

PONE-D-23-36838Protocol for the Swiss COhort of Healthcare Professionals and Informal CAregivers (SCOHPICA): professional trajectories, intention to stay in or leave the job and well-being of healthcare professionalsPLOS ONE

Dear Dr. Peytremann Bridevaux,

Thank you for submitting your manuscript to PLOS ONE. After careful consideration, we feel that it has merit but does not fully meet PLOS ONE’s publication criteria as it currently stands. Therefore, we invite you to submit a revised version of the manuscript that addresses the points raised during the review process.

We look forward to receiving your revised manuscript.

Kind regards,

Fatma Refaat Ahmed, Ph.D.

Academic Editor

PLOS ONE

2. PLOS requires an ORCID iD for the corresponding author in Editorial Manager on papers submitted after December 6th, 2016. Please ensure that you have an ORCID iD and that it is validated in Editorial Manager. To do this, go to ‘Update my Information’ (in the upper left-hand corner of the main menu), and click on the Fetch/Validate link next to the ORCID field. This will take you to the ORCID site and allow you to create a new iD or authenticate a pre-existing iD in Editorial Manager. Please see the following video for instructions on linking an ORCID iD to your Editorial Manager account: https://www.youtube.com/watch?v=_xcclfuvtxQ".

Reviewers' comments:

Reviewer's Responses to Questions

**Comments to the Author**

1. Does the manuscript provide a valid rationale for the proposed study, with clearly identified and justified research questions?

Reviewer #1: Yes

Reviewer #2: Yes

2. Is the protocol technically sound and planned in a manner that will lead to a meaningful outcome and allow testing the stated hypotheses?

Reviewer #1: Yes

Reviewer #2: Yes

3. Is the methodology feasible and described in sufficient detail to allow the work to be replicable?

Reviewer #1: Yes

Reviewer #2: Yes

4. Have the authors described where all data underlying the findings will be made available when the study is complete?

Reviewer #1: Yes

Reviewer #2: Yes

5. Is the manuscript presented in an intelligible fashion and written in standard English?

Reviewer #1: Yes

Reviewer #2: Yes

6. Review Comments to the Author

You may also provide optional suggestions and comments to authors that they might find helpful in planning their study.

Reviewer #1: This manuscript by Peytremann-Bridevaux and colleagues is a study protocol of SCOHPICA – a Swiss national prospective mixed-method observational study of the professional intention and well-being of healthcare professionals and informal caregivers. This is a critical study which has the potential to inform national policies, especially in view of challenges many countries are facing regarding workforce crisis.

Overall, this is a comprehensive protocol which is very well-written. Study methods and key outcome variables are adequately described, including the table overview. Ethical approval sought and detailed clearly. Looking forward to the results of this study.

Reviewer #2: Dear Authors,

Thank you for submitting a very interesting protocol, which stands very strong on its own. I have attached some comments to further increase its strength.

Good luck!

7. PLOS authors have the option to publish the peer review history of their article (what does this mean?). If published, this will include your full peer review and any attached files.

Reviewer #1: **Yes: **Kok Haw Jonathan Lim

Reviewer #2: **Yes: **Anke Boone

---

## [Author Response · Author response to Decision Letter 0]

3 May 2024

Response to the Reviewer #2 comments

Introduction

• It might be interesting to incorporate briefly into the introduction some of the European Union’s efforts to address job retention in healthcare, and to situate your research in this framework. For instance, the European Union has actively engaged in mitigating this matter by implementing multiple initiatives since 2012, such as the Joint Action Plan (JAP) for the EU Health Workforce (2013–2016) or establishment of the Support for the Health Workforce Planning and Forecasting Expert Network (SEPEN):

• Kroezen, M., Van Hoegaerden, M., & Batenburg, R. (2018). The Joint Action on Health Workforce Planning and Forecasting: Results of a European programme to improve health workforce policies. Health Policy, 122(2), 87–93. https://doi.org/10.1016/j.healthpol.2017.12.002

• Directorate-General for Health and Food Safety. (2023). Public Health Overview. https://health.ec.europa.eu/health-workforce/overview_en#sepen---support-for-the-health-workforce-planning-and-forecasting-expert-network-2017--2018

Our response: We thank the reviewer for this suggestion. We have now added these elements in the introduction; changes read as follows in the manuscript (line 72 to 81): 

“The issue of HCPs’ retention has been raised as a concern for over a decade (4). Since 2012, the European Union has launched several programs, such as the Join Action Plan (5) and the Support for the Health Workforce Planning and Forecasting Expert Network, (6) aimed at improving the retention of HCPs and addressing their shortage. In Switzerland, efforts have been made by health departments of the universities of applied sciences, which collaborated to create the Competence Network Health Workforce. This network aims to define a national strategy to tackle shortages of HCPs. At the political level, the Swiss population approved a constitutional law in 2021, compelling Swiss cantons and the federal government to ensure the sufficient availability of qualified nurses, and therefore to collect data to monitor the implementation of this new law (8). “

• You write ‘Life history calendar (LHC), a tool designed to collect retrospective data from participants by maximizing their possibilities of recalling past events completely and accurately (18, 19), has not been used in health workforce research yet, despite having been successfully used in different contexts.’ I am not familiar with this tool, but I am very interested. It might benefit the reader to briefly mention the contexts in which it has been successfully used.

Our response: We agree with the reviewer that it is indeed appropriate to mention some contexts in which LHCs have been used. We have added sentences to specify contexts in which LHCs have been applied successfully, changes appear in the text as follows (line 101 to 106): 

“LHCs have proven successful across various contexts, including studies on the trajectories of unemployed and vulnerable individuals, the sexual life of young people (25, 26), and in general population surveys (27). These studies have shown that 1) LHCs are more efficient than traditional sociodemographic questions for collecting retrospective data; 2) the data collected is reliable (28); and 3) online versions of LHCs can be used to reach large samples of participants (29, 30).”

• Consider adding a fourth area for further research, focusing on the inclusion of various healthcare settings (beyond hospitals or general practices), as this will clearly be a benefit from your study.

Our response: The suggested area has been added in the manuscript, and changes appear in lines 118, 119, as follows: “Finally, it is key to investigate settings beyond the two most frequently studied, namely hospitals and general practices. “

• Regarding the media's coverage of deteriorating working conditions and staff shortages among healthcare workers, it may be advisable to delete this information, as media reports may not always be reliable sources for such data.

Our response: We have removed the mention of media, as suggested.

• In the section about Switzerland, consider shortening the content and including more international studies on the topics to better address the article's international audience.

Our response: We have modified the text as proposed. Information about the Swiss context has been shortened and we have added information on the international context with the example of several countries. The whole section reads now as follows (line 121 to 144): 

“In Switzerland, the deteriorating working conditions of HCPs and staff shortages have been stressed by scientific studies and reports for several years, and this situation has worsened since the COVID-19 pandemic. Reports from the Swiss Health Observatory have predicted that a large number of HCPs would need to be hired to meet population needs, and that physicians’ supply in the ambulatory sector would not be sufficient by 2030 (42- 45). Additionally, a recent report has highlighted that 70,000 nursing staff will be needed by 2029, which encompasses both workforce replacements needs and the increased demand for additional staff stemming from population healthcare needs (46). In fact, the coverage rate is predicted to be lower than 80% with a clear deficit between workforce supply and projected needs (46, 47). This situation mirrors an international trend. Indeed, the WHO has projected a shortage of 15 million HCPs by 2030. In Germany, for example, estimates for the required number of HCPs in 2030 ranged from approximately 263,000 to nearly 500,000 full-time equivalents. (48) Similarly, as of September 2023, the UK’s National Health Service (NHS) reported 121,000 full-time equivalent vacant positions. (49) Finally, several reports have indicated that the United States will face a shortage of up to 124,000 physicians by 2033 and will require 200,000 nurses annually to meet the increasing care demand. (50) As in other countries, Swiss healthcare stakeholders have stressed the paucity of data, hindering effective monitoring, planning, and managing of the health workforce. Research projects aimed at both collecting data and leveraging HCPs retention have also been conducted in the Swiss context. These have investigated job stress, job satisfaction, burnout, and intention to leave the job/profession (51-65). However, like studies conducted in other countries, these publications mostly concentrated on nurses and physicians (mainly in hospital setting), and both nationwide and longitudinal data across multiple healthcare sectors are lacking to understand HCPs’ professional trajectories, well-being, and intention to stay in/leave their position/profession/health sector.” 

• The part concerning informal caregivers might be better situated within the study design rather than the introduction.

Our response: We agree with the reviewer; the section has been moved to the methods section (Study design sub-section, current lines 170 to 173). 

Methods and materials

• Sample size:

o Quantitative: ‘For the first baseline survey (fall 2022), 1500 HCPs were targeted.’ -> This was in fall 2022, this means you reached that target? And do you mean 5 to 10.000 unique participants? 

Our response: In the cohort, we target to follow at least 5000 to 10000 unique HCPs, in the medium term. As the cohort is an open cohort, which means that recruitment will be conducted every year, we expect this number to be achieved progressively. The target for the first recruitment was 1500. Upon reviewing the sentence you referenced, we acknowledge that mentioning the 1500 first participants may be unclear. Hence, the sentence has been clarified; it now reads as follows (line 189 to 190): 

“In the quantitative phase, SCOHPICA aims to collect data from 5,000 to 10,000 unique baseline HCPs, whom we will follow over the years.”

o Qualitative : 

‘Participants who agreed to be contacted in the baseline survey will be invited for individual or group sessions.’ In the event of a low response rate, will you continue open recruitment or adhere to those that agreed to be contacted in the baseline survey?

Our response: Thank you for this remark. Regardless of the response rate, participants included in the qualitative phase will be selected from the different recruitment conducted from 2022. Even a low response rate for the qualitative phase (for example 10%) would lead to include 500 HCPs in a qualitative study, considering the anticipated 5000 participants in SCOHPICA. This would be sufficient to identify participants for the qualitative phase of the study. 

What is your rationale for eight HCPs? Any reference or guidelines you are referring to?

Our response: There are no clear guidelines concerning the number of participants to include in concept mapping sessions and reviews are quite heterogeneous regarding this issue. Usually, concept mapping involves between 20 to 30 participants. However, some study with specific objectives have reported including fewer than 10 participants. It is well established that the number of participants is mainly linked to the characteristics of participants and the topic of the study, with careful consideration of the interaction between the two factors. Challenges in participant recruitment, as well as the expertise of participants regarding the topic of the study may justify smaller participant numbers in concept mapping sessions. In our case, concept mapping will be conducted with HCPs from different professions with varying work schedules, across different care contexts and cities throughout Switzerland. This makes it difficult to gather large numbers of HCPs for a single session. Therefore, we have pragmatically set the number of participants per session at eight. Although this number may be considered low compared to other studies using concept mapping, it will be compensated by conducing multiple concept mapping sessions, resulting in the inclusion of 120 participants in this qualitative phase. However, since not every aspect of the qualitative phase can be predetermined at this stage, we reserve the flexibility to adjust the number of participants as needed. This has been added in the article, which now reads as follows (lines 205-206): “according to the characteristics of participants, the chosen topic, the specific method and data saturation assessment”. 

We also added a reference (line 206). 

• Recruitment:

o Will recruitment efforts be continuous or periodic? When precisely will the survey be open each year: periodically or throughout the year?

Our response: The recruitment will be periodic and last four months every year. This has now been specified in the manuscript (lines 197 to 198): 

“Since SCOHPICA is an open cohort, new participants will be recruited every year between October 1st and January 31 of the following year.”

o Why limit the project presentation to Switzerland? Why not present it at international conferences? Many international conferences could offer a valuable platform for your study.

Our response: The presentations you mention are those planned for the recruitment of HCPs (section 2.2.2. recruitment of healthcare professionals). These are part of the communication strategy for the recruitment of participants, which is only conducted in Switzerland (i.e., inclusion criteria: working in Switzerland). 

• Data collection:

o Quantitative:

30 minutes is a long time, have you considered omitting some questions to avoid high dropout rates? Will there be any follow-up or reminders for people who do not complete the survey until the end? Additionally, if I understand correctly, participants do not have to register beforehand?

Our response: Actually, we have already reduced the initial length of the questionnaire to focus only on the determinants of the intent to stay most frequently reported in the literature. Sociodemographic and socio-professional questions comprise a large part of the baseline questionnaire, yet some of these will not be included in the follow-up questionnaire, resulting in shorter follow-up questionnaires. 

For the baseline questionnaire, we are not able to send reminders as we do not have HCPs’ email addresses. However, during the follow-up surveys, we will contact baseline participants who provided their contact details (i.e. email address), and send up to three reminders to those who did not respond to the questionnaire. 

Following data collection each year, we will assess missing data and consider shortening the questionnaire if we observe that some questions are systematically skipped, for example. 

I would also suggest to briefly mention which validated measurement scales will be used in the text (e.g. copsoq, etc), instead of only listening them in a table. The table can be used for detailed explanations and a comprehensive overview.

Our response: As suggested, we now mention the name of the scales in the manuscript and not only in Table 1. 

Changes appear as follows (lines 276 to 297): 

“We chose the following validated scales to measure these determinants (see details in Table 1): 

• Quantitative workload inventory (69): workload.

• Copenhagen Psychosocial Questionnaire (COPSOQ; 70): control over working time; possibilities for development; work-life conflict; influence at work; sense of community at work; meaning of work; job satisfaction.

• Practice Environment Scale of the Nursing Work Index (PES-NWI; 71): staffing and resources.

• Global transformational leadership scale (72): transformational leadership.

• Intensity of interprofessional collaboration (73): interprofessional practice.

• Recognition at work scale (74) : recognition from managers, colleagues and patients.

• Rushton moral resilience scale (75).

• Intolerance of uncertainty scale (76).

• The one-item MBI:EE (77): non-proprietary single-item of burnout.

• Self-rated health single item (78).

• Organizational Support Subscale of the Nursing Work Index (79): perceived quality of care.

Four not-validated items were drawn from a study of Shanafelt and colleagues (80) to complement the assessment of perceived quality of care. Moreover, two questions were developed by the research team to assess “Preparedness to work reality” (as no validated scale is currently available for this construct). We conducted a qualitative pre-test of the questionnaire with 20 participants from diverse backgrounds.”

Furthermore, I recommend clustering all the occupation-related aspects in the text to ensure a clear overview, while elaborating on them (i.e., all categories) in a table format. For example, mentioning all the different income categories in the manuscript is unnecessary; it would be more suitable to include them in a table or supplement.

Our response: The text has been shortened and details appear now in a new table (Table 2). Text now appears as follows (lines 298 to 310): 

“Additionally, the baseline questionnaire collects information on respondents’ work and occupation-related aspects (Table 2). 

Socio-professional information included: employment status, professional status, employment rate, occupational context, current profession, training and specialization, additional training, years of professional experience, main occupational domain, hours worked per day, days worked per week, type of work shift, type of work schedule, managerial responsibilities, country of education, other past profession(s), modification of employment rate in past 12 months, number of employers/occupational context/job location changes, since starting to work in healthcare, career interruption(s) of one year or longer, unemployment period(s), work-related accident(s)/sick leave(s), healthcare sector throughout career, healthcare sector of first employment, work location, and commuting time to work.

Sociodemographic information included: gender, year of birth, nationality, marital/partnership status, children, informal caregiving status, monthly individual incomes, monthly household incomes, and residency location.”

o Qualitative:

Will interviews and focus groups be conducted exclusively face-to-face, or will there be an option for online participation? Please discuss the advantages and disadvantages of both approaches and provide insight into the rationale behind your decisions.

o E.g. Boone, 

---

## [Decision Letter · Decision Letter 1]

19 Jul 2024

PONE-D-23-36838R1Protocol for the Swiss COhort of Healthcare Professionals and Informal CAregivers (SCOHPICA): professional trajectories, intention to stay in or leave the job and well-being of healthcare professionalsPLOS ONE

Dear Dr. Peytremann Bridevaux,

Thank you for submitting your manuscript to PLOS ONE. After careful consideration, we feel that it has merit but does not fully meet PLOS ONE’s publication criteria as it currently stands. Therefore, we invite you to submit a revised version of the manuscript that addresses the points raised during the review process.

We look forward to receiving your revised manuscript.

Kind regards,

Fatma Refaat Ahmed, Ph.D.

Academic Editor

PLOS ONE

Journal Requirements:

Reviewers' comments:

Reviewer's Responses to Questions

**Comments to the Author**

1. Does the manuscript provide a valid rationale for the proposed study, with clearly identified and justified research questions?

Reviewer #3: Yes

Reviewer #4: Yes

2. Is the protocol technically sound and planned in a manner that will lead to a meaningful outcome and allow testing the stated hypotheses?

Reviewer #3: Yes

Reviewer #4: Yes

3. Is the methodology feasible and described in sufficient detail to allow the work to be replicable?

Reviewer #3: Yes

Reviewer #4: Yes

4. Have the authors described where all data underlying the findings will be made available when the study is complete?

Reviewer #3: No

Reviewer #4: Yes

5. Is the manuscript presented in an intelligible fashion and written in standard English?

Reviewer #3: Yes

Reviewer #4: Yes

6. Review Comments to the Author

You may also provide optional suggestions and comments to authors that they might find helpful in planning their study.

Reviewer #3: The manuscript could be accepted after addressing minor comments. As of now, it looks good and authors sufficiently address comments of other reviewers in first stage of revision.

An abstract should contain a potential implications.

The keywords chosen are not particularly appropriate. It is recommended to rewrite the keywords based on the content of this article.

The methods section should include the design and setting of the study and a clear description of all processes, interventions and comparisons

Reviewer #4: Peytremann-Bridevaux et al. presented the manuscript study protocol Protocol for the Swiss COhort of Healthcare Professionals and Informal CAregivers (SCOHPICA): professional trajectories, intention to stay in or leave the job, and wellbeing of healthcare professionals. I’ve read the manuscript with great interest, the main messages are clear and could be interesting for the journal and a broader audience. The protocol fully describes all study procedures. It is a well-written manuscript and a well-prepared investigation.

7. PLOS authors have the option to publish the peer review history of their article (what does this mean?). If published, this will include your full peer review and any attached files.

Reviewer #3: No

Reviewer #4: No

---

## [Author Response · Author response to Decision Letter 1]

29 Jul 2024

Response to Reviewer #3:

We would like to thank the Reviewer for taking the time to review our protocol. Please find our responses to the comments below here. 

Reviewer #3: The manuscript could be accepted after addressing minor comments. As of now, it looks good and authors sufficiently address comments of other reviewers in first stage of revision. An abstract should contain a potential implications.

We agree that is it important to underscore the potential implications in the abstract. Therefore, we have made the following additions in bold. The modified Discussion section of the abstract now reads as follows:

Discussion: Using innovative methodologies, SCOHPICA will gather nationwide and longitudinal data on HCPs practicing in Switzerland. These data could have numerous implications: promoting the development of research related to HCPs’ well-being and retention intentions; supporting the development of policies to improve working conditions and career prospects; contributing to the evolution of training curricula for future or current healthcare professionals; aiding in the development of health systems capable of delivering quality care; and finally, providing the general public and stakeholders with free and open access to the study results through an online dashboard.

The keywords chosen are not particularly appropriate. It is recommended to rewrite the keywords based on the content of this article.

Many thanks for your feedback, we have amended the keywords as follows:

Healthcare professionals, career trajectories, well-being, retention intentions, staff turnover, cohort study, mixed methods

The methods section should include the design and setting of the study and a clear description of all processes, interventions and comparisons

We believe that the design and setting of the study, along with descriptions of all processes, measurements and analyzes, are thoroughly detailed in the Methods section of the manuscript. Please note that there are neither interventions nor specific comparisons in our study.

Response to Reviewer #4:

Reviewer #4: Peytremann-Bridevaux et al. presented the manuscript study protocol Protocol for the Swiss COhort of Healthcare Professionals and Informal CAregivers (SCOHPICA): professional trajectories, intention to stay in or leave the job, and wellbeing of healthcare professionals. I’ve read the manuscript with great interest, the main messages are clear and could be interesting for the journal and a broader audience. The protocol fully describes all study procedures. It is a well-written manuscript and a well-prepared investigation.

We thank the Reviewer for taking the time to review our manuscript and for the positive feedback, which is very encouraging. We are pleased to hear that that the Reviewer found the study protocol well-described and well-prepared.

---

## [Decision Letter · Decision Letter 2]

16 Aug 2024

Protocol for the Swiss COhort of Healthcare Professionals and Informal CAregivers (SCOHPICA): professional trajectories, intention to stay in or leave the job and well-being of healthcare professionals

PONE-D-23-36838R2

Dear Dr. Peytremann Bridevaux,

We’re pleased to inform you that your manuscript has been judged scientifically suitable for publication and will be formally accepted for publication once it meets all outstanding technical requirements.

Kind regards,

Fatma Refaat Ahmed, Ph.D.

Academic Editor

PLOS ONE

Additional Editor Comments (optional):

Reviewers' comments:

Reviewer's Responses to Questions

**Comments to the Author**

1. Does the manuscript provide a valid rationale for the proposed study, with clearly identified and justified research questions?

Reviewer #3: Yes

2. Is the protocol technically sound and planned in a manner that will lead to a meaningful outcome and allow testing the stated hypotheses?

Reviewer #3: Yes

3. Is the methodology feasible and described in sufficient detail to allow the work to be replicable?

Reviewer #3: Yes

4. Have the authors described where all data underlying the findings will be made available when the study is complete?

Reviewer #3: Yes

5. Is the manuscript presented in an intelligible fashion and written in standard English?

Reviewer #3: Yes

6. Review Comments to the Author

You may also provide optional suggestions and comments to authors that they might find helpful in planning their study.

Reviewer #3: My comments are sufficiently addressed in the revised manuscript. The editor makes the final decision.

7. PLOS authors have the option to publish the peer review history of their article (what does this mean?). If published, this will include your full peer review and any attached files.

Reviewer #3: No

---

## [Editor Report · Acceptance letter]

20 Aug 2024

PONE-D-23-36838R2 

PLOS ONE

Dear Dr. Peytremann-Bridevaux, 

I'm pleased to inform you that your manuscript has been deemed suitable for publication in PLOS ONE. Congratulations! Your manuscript is now being handed over to our production team.

Kind regards, 

on behalf of

Dr. Fatma Refaat Ahmed 

Academic Editor

PLOS ONE